# The Impacts of the Neighborhood Built Environment on Social Capital for Middle-Aged and Elderly Koreans

**Eunju Hwang [1],\*, Nancy Brossoie [2], Jin Wook Jeong [3] and Kimin Song [3]**

[1]  Department of Apparel, Housing and Resource Management, Virginia Tech, 295 West Campus Drive, Blacksburg, VA 24061, USA
[2]  Center for Gerontology, Virginia Tech, 230 Grove Lane, Blacksburg, VA 24061, USA; brossoie@vt.edu
[3]  Department of Medical and Digital Engineering, Hanyang University, 222 Wangsimni Seongdong-gu, Seoul 04763, Korea; jygin@hanyang.ac.kr (J.W.J.); haieung@hanyang.ac.kr (K.S.)
\*  Correspondence: hwange@vt.edu

**Abstract:** The purpose of this study was to investigate the relationship between the neighborhood built environment (NBE) aspects of age-friendly cities and communities (AFCCs) and social capital in the Korean context. We described and compared age differences when analyzing misfits of AFCC NBE and impacts on social capital. We collected the data ($N = 1246$) from two Korean communities; our multiple and binary logistic regression outcomes show that AFCC NBE aspects such as outdoor spaces, transportation, and housing are significant predictors of different subcategories of social capital. For the older group, the outdoor spaces misfit was significant for all three subcategories of social capital, but transportation and housing misfits were significant for the social trust and reciprocity index scores. For the middle-aged group, the outdoor spaces misfit was significant for social networking and participation, and a transportation misfit was significant for participation and social trust and reciprocity. Fewer misfits or better fits of outdoor spaces and transportation encouraged more networking, participation, social trust, and reciprocity. Dwelling type was important to predict social capital, especially for the older group. The present study confirmed the importance of AFCC NBE in predicting social capital and unique factors in the Korean context.

**Keywords:** built environment; walkability; social capital; age-friendly neighborhoods

## 1. Introduction

In a neighborhood context, social capital refers to the connections and resources made available through social interactions, contacts, networks, reciprocity, norms, and trust between neighbors [1]. The neighborhood built environment (NBE) includes features such as well-maintained public spaces, friendly neighbors, and safe streets, which collectively provide opportunities for connectivity and social capital [2]. While some NBE features such as aesthetics of street design, transit stops, sidewalk qualities, and street crossings [3] are important to building social capital for all age groups, some street-level NBE factors such as wayfinding aids, public restrooms, and benches create environments more comfortable for older adults [2]. Incorporating NBE features into communities encourages networking opportunities, social contacts, and participation, which are vital elements for building and maintaining social capital [4].

With the increasing emphasis on building suitable environments for growing older adult populations, a lot of attention has been given to how NBE contributes to their social connectedness and inclusion in the community. One example is the World Health Organization's Age-Friendly Cities and Communities (AFCCs) initiative, which is a comprehensive planning guide to implement accessible local infrastructure that is supportive of active aging [5]. The NBE aspects of AFCCs include the walkability and accessibility of outdoor spaces/buildings, the quality and utilization of public transportation, and the affordability and accessibility of housing [5]. The purpose of AFCCs is to promote an active living

environment and a connected society [5]. With the advent of AFCCs, many studies have focused on the impacts of NBE on health and quality of life [6–10]. However, there is a lack of research on how AFCC NBE features affect local residents' social capital [11]. Living in a community where the AFCC initiative is practiced is associated with better health and lower functional ability [5–7] and quality of life [8,9]. However, the relationship between AFCC NBE features and one's social capital, which is important for social connectivity, has not been studied. There is a lack of research on how AFCC NBE features affect local residents' social capital [11].

As one gets older, his/her activity space tends to shrink to their home and immediate neighborhood. Therefore, interactions between older users and their environment are critical and can be used to develop and maintain social capital [12]. These interactions between person and environment (P-E) in later life are explained by the congruence theory. Generally, when there is a good fit explaining the congruence between personal preferences and environmental characteristics, it is associated with a better quality of life, better health, and less stress [13]. However, when there is a poor P-E fit because of low-quality homes/buildings, broken windows, and unsupportive relationships with neighbors, this is associated with negative outcomes for both physical and mental health [10], and lower functional ability [6]. A lack of support and P-E misfits will create barriers. Thus, many communities have begun to address NBE aspects to remove barriers where P-E misfits are identified [10]. This can potentially improve social capital among residents. However, little is known about how AFCC NBE's P-E misfit affects social capital. To answer the question, we draw upon Putnam's definition of social capital [1], characterized as the accumulated physical and intellectual resources (actual and perceived) collected through social networks and exchanges.

Putnam described the acquisition of social capital through bonding or bridging. That is, social exchanges that occur within personal social networks (i.e., bonding) help meet members' needs through shared resources, which supports reliance on members and promotes group homogeneity. Conversely, exchanges that cross personal networks (i.e., bridging) help meet members' needs by drawing from diverse resources and establishing wider heterogeneous networks. One's ability and desire to engage with other community members and subsequently build social capital not only requires a shared sense of social norms and trust but also relies on environments that provide opportunities to engage.

We aim to investigate the impacts of AFCC NBE on social capital, specifically in the Korean context. The NBE literature has recently received more attention in Korea as many local Korean communities have started adopting AFCCs in their planning policies. As in Western countries, access to certain outdoor spaces (e.g., green space, places to exercise) [14], convenient public transportation [15], and well-maintained neighborhoods and housing [16] are related to social capital in Korea. Although these studies contributed to building more knowledge on NBE's impacts on social capital, the focus of these studies was on adolescents or adults in general, and they were conducted in a large city such as Seoul [14,16]. When the focus was on older adults' social capital as measured by NBE, access to senior-related local amenities [17], affordable public transportation options [15], and outdoor seating spaces [18] were identified as important factors related to social capital, especially for social networking. Other aspects of social capital such as social participation, trust, and reciprocity have not been fully explored [11,14]. Thus, the present study is interested in the relationship between AFCC NBE and social capital for older adults in the Korean context. Since different age groups perceive AFCC features significantly differently [19], the perceptions of older adults (aged 65 and older) and middle-aged adults (aged between 45 and 64) were utilized to analyze the incongruence or misfits of AFCC NBE P-E and their effect on social capital. First, we developed a P-E misfit index. Then, we investigated the associations between demographic and dwelling characteristics and the NBE P-E misfit index and analyzed the impacts of the index on social capital.

## 2. Materials and Methods

### 2.1. Sample and Data Collection

This study analyzed the data ($N$ = 1246) on AFCC NBE features collected in two communities located near mountains in South Korea. Community I is located in the central mountainous area in the southeast of Korea or North Gyeonsang Province, South Korea. The city is an urban-rural complex consisting of two Eup and seven Myeon (rural districts) and five Dong (urban districts) [20]. In 2019, community I had a population of 72,073 (population density: 79.01 per km$^2$) [21]. Community II is located in Kangwon Province in the northeast part of Korea. Like community I, the districts of community II also include an urban-rural complex consisting of one Eup and eight Myeon (rural districts) and 16 Dong (urban districts) [20]. In 2019, community II had a population of 352,860 (population density: 412 per km$^2$) [21]. Both communities have experienced an increasing number of older adults in the last decade. Community I had 28.9% older adults in 2019, an increase from 21.9% in 2010; community II had 14.3% older adults in 2019, an increase from 11.1% in 2010 [22]. Although the community size of the selected communities is different, they share similar geographic typology, being located near national parks and including many rural mountainous areas in their local districts. Additionally, both cities anticipate a steady increase in the number of older adults, with an associated decrease in the workforce until 2030 [23]. Because of this, both communities' priorities are to provide more supportive age-friendly environments and to build livable and connected neighborhoods led by local residents' associations. Both communities have started engaging local residents by connecting younger generations and older residents [24,25]. This makes them ideal sites to study how local residents perceive their NBE and its impacts on social capital to identify current resources and barriers.

A modified version of the United States' AARP Age-Friendly Neighborhood Livability Survey [26] was adopted, with questions with a community-level focus on walkability, accessibility, and age-friendliness. Two bilingual English-Korean speakers, including one teaching English as a second language, translated the survey questions. Residents ages 45 and older were invited to complete the survey so that the communities selected for this study could better understand the needs of current and future older residents. Local community centers and senior centers helped to locate potential study participants. Using a convenient sampling approach, the study was promoted at community and senior centers and local neighborhood activities. After excluding survey data from respondents younger than 45, 621 eligible surveys from community I and 625 eligible surveys from community II (a total of 1246 respondents) were used for the analyses. The data were collected using a self-administered questionnaire or a face-to-face interview for those who needed assistance filling out the questionnaire. The data collection followed the policies and procedures required by the ethics committee of Hanyang University (HYU-2019-03-005).

### 2.2. Measures

Social capital. Identifying and assessing social capital can be difficult because of its subjectivity, but it can be measured in several ways. The most common measurements of social capital in Korea include social networking, social participation, and trust [14]. Adopting this for our study, social capital was measured by three subscales, networking with neighbors, participation in neighborhood association activities, and social trust and reciprocity, as adopted from a previous urban neighborhood study [15]. Networking with neighbors was measured by a question asking, "In the past year, did you talk with your neighbors about a neighborhood issue or a problem?" To measure the participation in neighborhood association activities, we asked, "In the past year, did you attend a neighborhood organization meeting?" The responses to these two questions were coded as 1 = "yes" and 0 = "no" or "don't know".

Social trust and reciprocity were measured by the following three items: I feel respected by my neighbors. If my neighbors knew I needed help, they would be willing to help me. My neighbors know I am available to help them, if I know they need help.

Each question was measured using a five-point Likert scale, from 1 (strongly disagree) to 5 (strongly agree). A social trust and reciprocity index was developed by adding the response scores from each of the three items representing perceived social trust and reciprocity, which also represent the bridging activities identified by Putnam [1]. Summed scores range from 0 to 15. High scores indicate great trust and reciprocity between neighbors. The Cronbach's alpha coefficient for the three items was 0.79.

AFCC neighborhood built environment (NBE). Three subscales of AFCC NBE were created from the AARP's livability survey [26] on participants' perceptions of the availability and importance of three features in the built environment: outdoor spaces, transportation, and housing.

- Outdoor spaces were measured by 10 items related to adjacent land use and the pedestrian environment. These items assessed the availability of well-maintained and safe streets, parks, sidewalks, intersections, accessible public buildings and restrooms, benches in public spaces, and options for affordable parking.
- Public transportation features were measured by nine items related to public transportation and safety interventions. These items assessed the availability of accessible and affordable public transportation, traffic signs, speed limits, maintenance of transportation stops and public transportation vehicles, and special transportation for people with disabilities.
- Housing features were measured by two items related to affordability and accessibility of housing in one's neighborhood. These items assessed the availability of affordable housing options for older adults and accessible homes.

All questions started with "Does your neighborhood have . . . ?" For example, "Does your neighborhood have well-maintained and safe parks within walking distance of your home?" Each item was coded as 1 = "yes" or 0 = "no" or "don't know." Scores could range from 0 to 10 for the outdoor space index, from 0 to 9 for the transportation index, and from 0 to 2 for the housing index.

Person-environment (P-E) misfit indices. To create P-E misfit indices, we used Choi's approach [6], based on Lawton's P-E congruence theory. First, respondents' perceived availability and importance of the 21 NBE items was coded for each perception as 0 = not important/ not available and 1 = important/available. If a respondent perceived a feature as important but the feature was not available, the P-E misfit index for that item was coded as 1 = P-E, and a misfit exists. Index scores ranged from 0 to 21. A higher number reflected a greater P-E misfit. The Cronbach's alpha coefficient was 0.685 for the outdoor spaces misfit index, 0.631 for the transportation misfit index, and 0.611 for the housing misfit index.

Demographic and dwelling characteristics included: age (in years); gender (1 = female, 0 = male); marital status (1 = married, 0 = other); education level (1 = elementary school to 4 = graduate school); annual household income (1: <USD 10,000 to 5: >USD 70,000); self-rated health (1 = very poor to 5 = excellent); dwelling type (1 = single-family home, 0 = other), home ownership (1 = homeowner, 0 = renter), and length of residence in the current neighborhood (1 = less than a year to 4 = more than 15 years).

### 2.3. Data Analysis

To provide an overview of sample characteristics, first, we conducted a descriptive analysis. Data were described using means and standard deviations or frequencies and proportions, as appropriate. Age differences in demographic and dwelling characteristics were analyzed using *t*-tests for continuous variables or chi-square tests for categorical variables as appropriate. Then, bivariate analyses were conducted to evaluate the correlations between key variables. Lastly, multivariate analyses were conducted to analyze the relationships between NBE and social capital. A multilevel approach using social capital as an outcome to identify age differences was suggested in a previous AFCC study [11]. To ensure we were measuring discrete concepts, we assessed the item multicollinearity by checking the variance inflation factor (VIF). All VIF scores were between 1.072 and 1.669, which was within the acceptable range of less than four [27]. Logistic and linear regression

models were developed using the IBM SPSS version 26.0 statistical software, Chicago, IL, the United States, with *p*-values < 0.05 set as the threshold for statistical significance.

## 3. Results

### 3.1. Characteristics of the Participants

Table 1 presents descriptive information on the sample. Among the 1246 participants, older adults (65 and older) accounted for 658 respondents and middle-aged respondents (ages 45 to 64) accounted for 588. The participants' age ranged from 45 to 96. The mean age was 74.61 years (SD = 6.26) in the older group and 55.66 years (SD = 5.53) in the middle-aged group. The older group comprised a lower proportion of married status ($\chi^2$ = 74.30, $p < 0.001$), but had a higher proportion of elementary or middle school education ($t = -19.13$, $p < 0.001$) and annual household income between 0 and 29,999,999 KRW ($t = -20.58$, $p < 0.001$) than the middle-aged group. However, older adults rated their health ($t = -6.64$, $p < 0.001$) more positively than the middle-aged group did. The proportions of older adults living in single-family homes ($\chi^2$ = 42.41, $p < 0.001$) and owners ($\chi^2$ = 7.73, $p < 0.01$) were higher than in the middle-aged group. Older adults' length of residence in the current neighborhood was higher than for the middle-aged group ($t = 2.23$, $p < 0.001$).

**Table 1.** Characteristics of the participants ($N$ = 1246).

| | Mean or Frequency (%) | | $\chi^2$ or *t*-Values |
|---|---|---|---|
| | **Older Group (65+ Years)** **($n$ = 658)** | **Middle-Aged Group (45–64 Years)** **($n$ = 588)** | |
| **Demographic Characteristics** | | | |
| Age (Range: 45–96) | 74.61 (SD: 6.26) | 55.66 (SD: 5.53) | 56.66 *** |
| Gender | | | 0.205 |
| Female | 357 (54.3) | 340 (57.8) | |
| Male | 301 (45.7) | 248 (42.2) | |
| Marital Status (Married) | 451 (68.5) | 522 (88.2) | 74.30 *** |
| Education | | | −19.13 *** |
| Elementary or middle school | 360 (54.7) | 59 (10.0) | |
| High school | 277 (42.1) | 424 (72.1) | |
| University/college | 20 (3.0) | 103 (17.5) | |
| Graduate school | 1 (0.2) | 2 (3.4) | |
| Annual Household Income | | | −20.58 *** |
| 0–9,999,999 KRW (approx. 0–9999 USD) | 304 (46.2) | 51 (8.7) | |
| 10,000,000–29,999,999 KRW (10,000–29,999 USD) | 274 (41.6) | 213 (36.2) | |
| 30,000,000–49,999,999 KRW (30,000–49,999 USD) | 73 (11.1) | 266 (45.2) | |
| 50,000–69,999,999 KRW (50,000–69,999 USD) | 5 (0.8) | 53 (9.0) | |
| >70,000,000 KRW (>70,000 USD) | 2 (0.3) | 5 (0.9) | |
| Self-Rated Health | | | 6.64 *** |
| Poor | 27 (4.1) | 50 (8.5) | |
| Fair | 311 (47.3) | 317 (54.0) | |
| Good | 241 (36.6) | 189 (32.1) | |
| Very good | 76 (11.5) | 32 (5.4) | |
| Excellent | 3 (0.5) | 0 (0) | |
| **Dwelling Characteristics** | | | |
| Dwelling Type | | | 42.41 *** |
| Single-family home | 335 (50.9) | 192 (32.7) | |
| Other (e.g., condominium/apartment, townhouse, etc.) | 323 (49.1) | 396 (67.3) | |
| Tenure | | | 7.73 ** |
| Owner | 517 (78.6) | 422 (71.8) | |
| Renter | 141 (21.4) | 166 (27.2) | |
| Length of Residence in Current Neighborhood | | | 2.23 *** |
| Less than 1 year | 7 (1.1) | 11 (1.9) | |
| 1–5 years | 45 (6.8) | 30 (5.1) | |
| 6–15 years | 109 (16.6) | 153 (26.0) | |
| >15 years | 497 (75.5) | 394 (67.0) | |

** $p < 0.01$; *** $p < 0.001$.

### 3.2. Frequency Analysis

Table 2 summarizes a frequency analysis of participants' perceived neighborhood built environment (NBE) availability, NBE misfits, and social capital. Regarding the perception of NBE availability, overall, both the older and middle-aged groups perceived transportation as most available, and housing-related items as least available. However, the total mean of transportation was not significant, while the total mean of housing was significant ($t = -6.39$, $p < 0.001$) between the older and middle-aged groups. The middle-aged group was likely to perceive a greater availability of housing-related items.

**Table 2.** Frequency analysis.

| Features | Survey Items | Older Group (*n* = 658) | | Middle-Aged Group (*n* = 588) | |
|---|---|---|---|---|---|
| | Perception of NBE Availability | | | | |
| | | Yes Available (%) | No or Not Sure (%) | Yes Available (%) | No or Not Sure (%) |
| Outdoor spaces (range: 1–10) | Well-maintained and safe parks within walking distance of your home | 72.8 | 27.2 | 76.0 | 24.0 |
| | Public spaces with enough benches | 54.9 | 45.1 | 36.7 | 63.3 |
| | Sidewalks in good condition, free from obstruction, and are safe for pedestrian use and accessible for wheelchairs or other assistive mobility devices | 55.3 | 44.7 | 70.6 | 29.4 |
| | Separate pathways for bicycles and pedestrians | 21.6 | 78.4 | 27.2 | 72.8 |
| | Well-maintained public buildings and facilities that are accessible to people of different physical abilities | 46.5 | 53.5 | 53.1 | 46.9 |
| | Well-maintained public restrooms that are accessible to people of different physical abilities | 31.8 | 68.2 | 36.9 | 63.1 |
| | Well-maintained streets | 82.7 | 17.3 | 83.7 | 16.3 |
| | Public parking lots/areas to park | 52.1 | 47.9 | 53.2 | 46.8 |
| | Affordable public parking | 43.0 | 57.0 | 39.6 | 60.4 |
| | Well-lit, safe streets, and intersections. | 65.3 | 34.7 | 71.6 | 28.4 |
| | Total measured mean | 5.25 | | 5.48 *t*-value = −1.79 | |
| Transportation (range: 1–9) | Accessible and convenient public transportation | 85.6 | 14.4 | 90.5 | 9.5 |
| | Affordable public transportation | 71.7 | 28.3 | 68.4 | 31.6 |
| | Well-maintained public transportation vehicles | 75.1 | 24.9 | 65.0 | 35.0 |
| | Reliable public transportation | 73.7 | 26.3 | 69.4 | 30.6 |
| | Safe public transportation stops or areas | 75.8 | 24.2 | 72.6 | 27.4 |
| | Special transportation services for people with disabilities and older adults | 48.9 | 51.1 | 42.9 | 57.1 |
| | Easy to read traffic sign | 77.2 | 22.8 | 83.8 | 16.2 |
| | Enforced speed limits | 68.4 | 31.6 | 66.3 | 33.7 |
| | Audio/visual signals at pedestrian crossings | 33.9 | 66.1 | 30.1 | 69.9 |
| | Total measured mean | 6.10 | | 5.88 *t*-value = 1.73 | |
| Housing (range: 1–2) | Affordable housing options for older adults with shared facilities and outdoor spaces | 37.8 | 62.2 | 53.9 | 46.1 |
| | Homes equipped with accessible features (e.g., no-step entry, wider doorways, first-floor bedroom, and bathroom, grab bars) | 31.2 | 68.8 | 43.7 | 56.3 |
| | Total measured mean | 0.69 | | 0.96 *t*-value = −6.39 *** | |

**Table 2.** *Cont.*

| Features | Survey Items | Older Group (n = 658) | Middle-Aged Group (n = 588) |
|---|---|---|---|
| | NBE Misfit Indices | | |
| Outdoor spaces (range: 0–10) | | 1.72 | 1.22 t-value = 5.04 *** |
| Transportation (range: 0–9) | | 0.72 | 0.79 t-value = −1.01 |
| Housing (range: 0–2) | | 0.81 | 0.47 t-value = 7.73 *** |
| | Social Capital | | |
| Networking with neighbors (binary, at least once) | | 368 (55.9) | 429 (73.0) $\chi^2$ = 39.08 *** |
| Participation in neighborhood association activities(binary, at least once) | | 260 (39.5) | 219 (37.2) $\chi^2$ = 0.675 |
| Social trust and reciprocity index (range 5–15) | | 10.67 | 10.45 t-value = 2.12 * |

* $p < 0.05$; *** $p < 0.001$. NBE = Neighborhood built environment.

In terms of NBE misfits, there were significant differences in terms of outdoor spaces and housing. The older groups' mean scores for outdoor spaces and housing were significantly higher than the middle-aged group's ($t = 5.04$, $p < 0.001$; $t = 7.73$, $p < 0.001$).

Two subcategories of social capital were significantly different between the older and middle-aged groups. The older group's networking with neighbors was significantly lower than in the middle-aged group ($\chi^2 = 39.08$, $p < 0.001$), while the social trust and reciprocity index was significantly higher than in the middle-aged group ($t = 2.12$, $p < 0.05$).

*3.3. Bivariate Results*

Table 3 presents the pairwise correlation analysis results among key variables. There was a significant positive correlation between outdoor spaces and transportation misfit indices ($r = 0.519$, $p < 0.001$), and between outdoor spaces and housing misfit indices ($r = 0.439$, $p < 0.001$). However, correlations between the outdoor spaces misfit index and all social capital subscales of networking with neighbors, participation in neighborhood association activities, and social trust and reciprocity index scores were significant but negative ($r = −0.231$, $p < 0.001$; $r = −0.157$, $p < 0.001$; $r = −0.201$, $p < 0.001$, respectively). Correlations between transportation and housing misfit indices were significant and positive ($r = 0.210$, $p < 0.001$), but were negative with all three social capital subscales of networking with neighbors, participation in neighborhood association activities, and social trust and reciprocity index scores ($r = −0.125$, $p < 0.001$; $r = −0.197$, $p < 0.001$; $r = −0.237$, $p < 0.001$ respectively). The housing misfit index was negatively correlated with all three social capital subscales as well networking with neighbors, participation in neighborhood association activities, and social trust and reciprocity index scores ($r = −0.074$, $p < 0.001$; $r = −0.011$, $p < 0.001$; $r = −0.045$, $p < 0.001$ respectively). On the other hand, networking with neighbors and participation in neighborhood association activity were significantly and positively correlated ($r = 0.387$, $p < 0.001$) and networking with neighbors and social trust and reciprocity index scores were also positively correlated ($r = 0.347$, $p < 0.001$). There was a positive significant correlation between neighborhood association activity and social trust and reciprocity index scores ($r = 0.358$, $p < 0.001$).

**Table 3.** Pearson correlation among key variables.

| Variables | 1 | 2 | 3 | 4 | 5 | 6 |
|---|---|---|---|---|---|---|
| 1. Outdoor spaces misfit index | 1 | | | | | |
| 2. Transportation misfit index | 0.519 *** | 1 | | | | |
| 3. Housing misfit index | 0.439 *** | 0.210 *** | 1 | | | |
| 4. Networking with neighbors | −0.231 *** | −0.125 *** | −0.074 *** | 1 | | |
| 5. Participating in neighborhood association activities | −0.157 *** | −0.197 *** | −0.011 | 0.387 *** | 1 | |
| 6. Social trust and reciprocity index | −0.201 *** | −0.237 *** | −0.045 | 0.347 *** | 0.358 *** | 1 |

*** Correlation is significant at the 0.001 level (two-tailed).

### 3.4. Multivariate Results

3.4.1. Predictors of Neighborhood Built Environment Misfit Indices

Table 4 presents regression models of AFCC NBE misfit indices by age groups. Among older respondents, homeownership and household income were the most important predictors of NBE misfits. Homeownership and household income had negative relationships with all indices. Among older respondents who did not own a home (e.g., renters), there were greater NBE misfits with outdoor spaces ($\beta = -0.211$, $p < 0.001$), transportation ($\beta = -0.096$, $p < 0.05$), and housing ($\beta = -0.210$, $p < 0.001$). When household income was low, greater NBE misfits in outdoor spaces ($\beta = -0.100$, $p < 0.05$), transportation ($\beta = -0.084$, $p < 0.05$), and housing ($\beta = -0.083$, $p < 0.05$) emerged. Age and length of residence in one's current neighborhood also had negative impacts on two subscales of NBE, outdoor spaces and transportation. Gender, marital status, and dwelling type were significant on different subscales of NBE misfits. Men and those who lived in single-family dwellings were more likely to perceive a housing misfit ($\beta = -0.091$, $p < 0.05$; $\beta = 0.095$, $p < 0.05$ respectively). Moreover, respondents who were not married were likely to experience a greater outdoor spaces misfit ($\beta = -0.105$, $p < 0.01$).

**Table 4.** Predictors of neighborhood built environment misfit indices by age groups.

| | Older Age Group | | | | | | Middle-Aged Group | | | | | |
|---|---|---|---|---|---|---|---|---|---|---|---|---|
| | Outdoor Spaces Misfit Index | | Transportation Misfit Index | | Housing Misfit Index | | Outdoor Spaces Misfit Index | | Transportation Misfit Index | | Housing Misfit Index | |
| | B | β | B | β | B | β | B | β | B | β | B | β |
| Age | −0.043 *** | −0.142 | −0.020 * | −0.101 | −0.004 | −0.033 | 0.049 *** | 0.177 | 0.008 | 0.039 | 0.026 *** | 0.197 |
| Gender (female) | 0.012 | 0.003 | 0.044 | 0.017 | −0.152 * | −0.091 | 0.447 *** | 0.143 | 0.285 ** | 0.122 | 0.117 * | 0.080 |
| Marital status (married) | −0.431 ** | −0.105 | −0.157 | −0.058 | −0.077 | −0.043 | 0.006 | 0.001 | 0.316 * | 0.087 | 0.019 | 0.008 |
| Education | 0.191 | 0.056 | 0.186 | 0.084 | −0.047 | −0.032 | −0.052 * | −0.018 | −0.030 | −0.014 | −0.023 | −0.017 |
| Household income | −0.265 * | −0.100 | −0.146 * | −0.084 | −0.095 * | −0.083 | −0.199 | −0.104 | 0.020 | 0.014 | −0.084 * | −0.094 |
| Self-rated health | −0.062 | −0.025 | 0.076 | 0.046 | −0.026 | −0.024 | −0.162 | −0.075 | −0.107 | −0.066 | −0.025 | −0.025 |
| Dwelling type (single-family home) | −0.075 | −0.020 | −0.080 | −0.032 | 0.158 * | 0.095 | −0.192 | −0.059 | −0.159 | −0.065 | −0.133 * | −0.087 |
| Length of residence | −0.199 * | −0.068 | −0.154 * | −0.080 | −0.020 | −0.016 | −0.131 | −0.058 | −0.075 | −0.044 | −0.015 | −0.014 |
| Home ownership (owner) | −0.987 *** | −0.211 | −0.295 * | −0.096 | −0.427 *** | −0.210 | −0.550 *** | −0.161 | −0.424 *** | −0.166 | −0.109 | −0.069 |
| Constant | 7.118 *** | | 2.943 *** | | 1.903 *** | | 0.150 | | 0.788 | | −0.535 | |
| $R^2$ | 0.109 | | 0.053 | | 0.059 *** | | 0.115 | | 0.060 | | 0.077 | |
| F | 8.805 *** | | 4.058 *** | | 4.541 *** | | 8.312 *** | | 4.122 *** | | 5.334 *** | |

\* $p < 0.05$; ** $p < 0.01$; *** $p < 0.001$.

Among middle-aged respondents, gender was the most important predictor of all three NBE misfit indices. Women were significantly more likely to experience a misfit with outdoor spaces ($\beta = 0.143$, $p < 0.001$), transportation ($\beta = 0.122$, $p < 0.01$), and housing ($\beta = 0.080$, $p < 0.05$) than men. Unlike the older group, middle age was associated positively with outdoor spaces ($\beta = 0.177$, $p < 0.001$) and housing ($\beta = 0.197$, $p < 0.001$). However, similar to the older group, among respondents who did not own a home, NBE misfits existed with outdoor spaces ($\beta = -0.161$, $p < 0.001$) and transportation ($\beta = -0.166$, $p < 0.001$). In

contrast, household income and living in single-family dwelling were significant only in predicting housing misfits for the middle-aged group ($\beta = -0.094$, $p < 0.05$; $\beta = -0.087$, $p < 0.05$). Length of residence was not significant for the middle-aged group.

### 3.4.2. Predictors of Social Capital

To identify predictors of three subcategories of social capital, binary logistic and multiple regression analyses were conducted with demographic variables and NBE misfit indices. The results of binary logistic and multiple linear regression models show that AFCC NBE aspects such as outdoor spaces, transportation, and housing were significant predictors of social capital.

Table 5 shows the results of a binary logistic regression on networking with neighbors. For both age groups, the outdoor spaces misfit index regarding the person-environment misfit was important. In the older group, demographic and dwelling characteristics (i.e., age, marital status, dwelling type, and length of residence in the current neighborhood) emerged as significant predictors. Specifically, respondents who were younger, not married, did not live in a single-family home dwelling type (e.g., condominium, apartment, townhouse), had a longer length of residence, and had a low outdoor space misfit index had a higher probability of networking with neighbors. Of the three NBE indices, only the outdoor spaces misfit index was significant. Collectively, they explained 22% of the variance in predicting networking with neighbors.

**Table 5.** Predictors of networking with neighbors by age groups.

| Variables | Older Age Group | | | | Middle-Aged Group | | | |
|---|---|---|---|---|---|---|---|---|
| | | | 95% CI for OR | | | | 95% CI for OR | |
| | B | OR | Lower | Upper | B | OR | Lower | Upper |
| Age | −0.036 * | 0.964 | 0.933 | 0.997 | −0.011 | 0.989 | 0.953 | 1.027 |
| Gender (female) | 0.110 | 1.116 | 0.782 | 1.594 | 0.003 | 1.003 | 0.670 | 1.500 |
| Marital status (married) | −0.429 * | 0.651 | 0.443 | 0.956 | −1.168 *** | 0.311 | 0.174 | 0.554 |
| Education | −0.246 | 0.782 | 0.549 | 1.114 | 0.129 | 1.138 | 0.765 | 1.693 |
| Household income | 0.228 | 1.256 | 0.967 | 1.632 | −0.258 | 0.772 | 0.592 | 1.008 |
| Self-rated health | −0.073 | 0.930 | 0.741 | 1.167 | −0.596 *** | 0.551 | 0.417 | 0.727 |
| Dwelling type (single-family home) | −1.199 *** | 0.301 | 0.210 | 0.432 | −0.100 | 0.905 | 0.579 | 1.413 |
| Length of residence | 0.343 * | 1.409 | 1.074 | 1.850 | 0.367 * | 1.444 | 1.089 | 1.914 |
| Tenure (owner) | 0.363 | 1.437 | 0.906 | 2.280 | −0.557 * | 0.573 | 0.369 | 0.889 |
| Outdoor spaces misfit index | −0.906 *** | 0.404 | 0.271 | 0.602 | −0.752 *** | 0.471 | 0.300 | 0.741 |
| Transportation misfit index | −0.350 | 0.705 | 0.471 | 1.054 | 0.045 | 1.046 | 0.681 | 1.605 |
| Housing misfit index | 0.195 | 1.215 | 0.967 | 1.527 | −0.031 | 0.969 | 0.727 | 1.292 |
| Constant | 2.854 | 17.349 | | | 2.758 | 15.765 | | |
| −2 Log likelihood | | 783.647 *** | | | | 616.269 *** | | |
| Nagelkerke $R^2$ | | 0.222 | | | | 0.163 | | |
| Goodness of fit | | $X^2 = 8.010$ | | | | $X^2 = 6.980$ | | |
| (Homer and Lemeshow Test) | | Sig = 0.433 | | | | Sig 0.539 | | |

\* $p < 0.05$; \*\*\* $p < 0.001$. CI = Confidence Interval. OR = Odds Ratio. Dependent variable: 1 (talked with neighbors at least once in the past year) or 0 (none). Education ranged from 1 to 4 (elementary to graduate school). Household income was measured by annual household income and ranged from 1 to 5 (<USD 10,000 to >USD 70,000). Self-rated health ranged from 1 to 5 (very poor to excellent). Length of residence ranged from 1 to 4 (<1 year to >15 years). Variables were entered in two steps. In step 1, demographic and dwelling characteristics were entered; in step 2, the P-E indices were entered. Table 5 presents the outcome of step 2 only.

Similarly, in the middle-aged group, not being married, a long length of residency, and low outdoor spaces misfit index scores were related to a higher probability to network with neighbors. However, unlike in the older group, self-rated health and homeownership were also related to a higher probability of networking with one's neighbors. The outdoor spaces misfit index was also significant. Together, the variables explained 16% of the variance in predicting networking with neighbors.

To examine the model fit, we conducted a Hosmer–Lemeshow test. According to this test, a poor fit is indicated by a significance value lower than 0.05 [28]. The chi-square

values for the Hosmer–Lemeshow test of the older group were 8.010 with a significance level of 0.433 and 6.980 with a significance level of 0.539 for the middle-aged group. These values are larger than 0.05, thus indicating support for the models.

Table 6 shows the results of a binary logistic regression on predicting social participation in neighborhood activities. Demographic variables, dwelling characteristics, and NBE misfit indices were entered to predict social participation. In the older group, two demographic and dwelling characteristics (i.e., dwelling type and length of residence) and outdoor spaces misfit index predicted participation (Nagelkerke $R^2$ = 0.182). For the middle-aged group, the outdoor spaces and transportation misfit indices were significant predictors as were age and gender (Nagelkerke $R^2$ = 0.150). Again, for both age groups, the outdoor spaces misfit index regarding the person-environment misfit was an important variable in predicting the participation of neighborhood organization activities. The chi-square value for the Hosmer–Lemeshow test of the older group was 5.674 with a significance level of 0.684 and 7.015 with a significance level of 0.535 for the middle-aged group. These values are larger than 0.05, thus indicating support for the models.

**Table 6.** Predictors of social participation in neighborhood activities by age groups.

| Variables | Older Group | | | | Middle-Aged Group | | | |
|---|---|---|---|---|---|---|---|---|
| | | | 95% CI for OR | | | | 95% CI for OR | |
| | B | OR | Lower | Upper | B | OR | Lower | Upper |
| Age | −0.022 | 0.979 | 0.948 | 1.011 | 0.053 ** | 1.055 | 1.018 | 1.093 |
| Age | −0.138 | 0.871 | 0.611 | 1.241 | 0.397 * | 1.488 | 1.030 | 2.148 |
| Gender (female) | −0.314 | 0.730 | 0.493 | 1.083 | −0.390 | 0.677 | 0.371 | 1.237 |
| Marital status (married) | −0.195 | 0.823 | 0.577 | 1.174 | −0.169 | 0.844 | 0.587 | 1.214 |
| Education | 0.124 | 1.132 | 0.878 | 1.459 | −0.092 | 0.912 | 0.714 | 1.166 |
| Household income | −0.036 | 0.965 | 0.772 | 1.206 | 0.043 | 1.043 | 0.809 | 1.346 |
| Self-rated health | −1.165 *** | 0.312 | 0.218 | 0.446 | −0.318 | 0.728 | 0.495 | 1.071 |
| Dwelling type (single-family home) | 0.382 * | 1.466 | 1.087 | 1.978 | 0.281 | 1.325 | 0.997 | 1.761 |
| Length of residence | −0.113 | 0.893 | 0.550 | 1.452 | −0.048 | 0.953 | 0.619 | 1.467 |
| Outdoor spaces misfit index | −0.468 * | 0.626 | 0.419 | 0.934 | −0.580 ** | 0.560 | 0.360 | 0.872 |
| Transportation misfit index | −0.301 | 0.740 | 0.493 | 1.113 | −0.837 *** | 0.433 | 0.293 | 0.639 |
| Housing misfit index | 0.220 | 1.247 | 0.999 | 1.555 | −0.068 | 0.934 | 0.710 | 1.229 |
| Constant | 0.775 | 2.171 | | | −3.372 | 0.034 | | |
| −2 Log likelihood | 788.057 *** | | | | 708.032 *** | | | |
| Nagelkerke $R^2$ | 0.182 | | | | 0.150 | | | |
| Goodness fit (Homer-Lemeshow Test) | $X^2$ = 5.674 (Sig = 0.684) | | | | $X^2$ = 7.015 (Sig = 0.535) | | | |

* $p < 0.05$; ** $p < 0.01$; *** $p < 0.001$. CI = Confidence Interval. OR = Odds Ratio. Dependent variable: 1 (participated in neighborhood organization meetings at least once in the past year) or 0 (none). Education ranged from 1 to 4 (elementary to graduate school). Household income was measured by annual household income and ranged from 1 to 5 (<USD 10,000 to >USD 70,000). Self-rated health ranged from 1 to 5 (very poor to excellent). Length of residence ranged from 1 to 4 (<1 year to >15 years). Variables were entered in two steps. In step 1, demographic and dwelling characteristics were entered; in step 2, the P-E indices were entered. Table 6 presents the outcome of step 2 only.

Lastly, a multiple linear regression analysis was conducted to predict the social trust and reciprocity index scores using demographic variables, dwelling characteristics, and NBE misfit indices (Table 7). In the older group, education, dwelling type, and all three NBE misfit indices were significant. Those who were less educated (β = −0.127, $p < 0.01$), living in single-family homes (β = 0.205, $p < 0.001$), with low misfit index scores in outdoor spaces (β = −0.168, $p < 0.001$) and transportation (β = −0.180, $p < 0.001$), and a high misfit index score in housing (β = 0.077, $p < 0.01$) were likely to have high social trust and reciprocity index scores. In the middle-aged group, variables including gender, marital status, self-rated health, dwelling type, and length of residence were significant predictors. However, only the transportation misfit index was significant. Specifically, being male (β = −0.086, $p < 0.05$), married (β = 0.179, $p < 0.001$), low self-rated health (β = −0.114, $p < 0.01$), living in single-family homes (β = 0.112, $p < 0.01$), long length of residence (β = 0.142, $p < 0.001$) in the current neighborhood, and low transportation misfit index score (β = −0.138, $p < 0.01$) significantly predicted social trust and reciprocity index scores. Overall, the regression

models explained 19.4% of the variance for the older group and 14.3% of the variance for the middle-aged group.

**Table 7.** Predictors of social trust and reciprocity index by age groups.

| Variables | Older Group | | | | Middle-Aged Group | | | |
|---|---|---|---|---|---|---|---|---|
| | | | 95% CI for B | | | | 95% CI for B | |
| | **B** | **β** | **Lower** | **Upper** | **B** | **β** | **Lower** | **Upper** |
| Constant | 11.751 *** | | | | 9.087 *** | | | |
| Age | −0.017 | −0.056 | −0.042 | 0.008 | 0.014 | 0.043 | −0.013 | 0.040 |
| Gender (female) | −0.153 | −0.040 | −0.430 | 0.123 | −0.308 * | −0.086 | −0.588 | −0.028 |
| Marital status (married) | 0.280 | 0.069 | −0.023 | 0.584 | 1.002 *** | 0.179 | 0.560 | 1.445 |
| Education | −0.425 ** | −0.127 | −0.700 | −0.150 | −0.184 | −0.055 | −0.456 | 0.088 |
| Household income | 0.067 | 0.025 | −0.135 | 0.268 | −0.174 | −0.079 | −0.360 | 0.012 |
| Self-rated health | −0.052 | −0.021 | −0.228 | 0.125 | −0.283 ** | −0.114 | −0.476 | −0.091 |
| Dwelling type (single-family home) | 0.775 *** | 0.205 | 0.495 | 1.056 | 0.422 ** | 0.112 | 0.122 | 0.723 |
| Length of residence | 0.163 | 0.056 | −0.047 | 0.373 | 0.373 *** | 0.142 | 0.165 | 0.580 |
| Tenure (owner) | 0.104 | 0.023 | −0.251 | 0.459 | 0.190 | 0.048 | −0.130 | 0.510 |
| Outdoor spaces misfit index | −0.166 *** | −0.168 | −0.255 | −0.078 | 0.026 | 0.022 | −0.091 | 0.142 |
| Transportation misfit index | −0.271 *** | −0.180 | −0.395 | −0.147 | −0.213 ** | −0.138 | −0.353 | −0.072 |
| Housing misfit index | 0.176 * | 0.077 | 0.000 | 0.351 | −0.190 | −0.077 | −0.405 | 0.025 |
| $R^2$ | 0.194 | | | | 0.143 | | | |
| F | 12.922 *** | | | | 8.009 *** | | | |

\* $p < 0.05$; \*\* $p < 0.01$; \*\*\* $p < 0.001$. CI = Confidence Interval. Dependent variable: Social trust and reciprocity index score ranged from 1 to 15. Education ranged from 1 to 4 (elementary to graduate school). Household income was measured by annual household income and ranged from 1 to 5 (<USD 10,000 to >USD 70,000). Self-rated health ranged from 1 to 5 (very poor to excellent). Length of residence ranged from 1 to 4 (<1 year to >15 years). Variables were entered in two steps. In step 1, demographic and dwelling characteristics were entered; in step 2, the P-E indices were entered. Table 7 presents the outcome of step 2 only.

## 4. Discussion

The focus of this study was to investigate the impacts of age-friendly cities and communities (AFCCs) neighborhood built environment (NBE) misfits on social capital. Overall, the NBE misfit indices were significant predictors of the aspects of social capital surveyed, with some similarities and differences between the older and middle-aged groups. For older adults, all three NBE misfit indices (outdoor spaces, transportation, and housing) emerged as significant predictors of social capital. Among the middle-aged group, only outdoor spaces and transportation were significant predictors.

The outdoor spaces misfit index was the most consistent predictor of social capital for the older group as it predicted networking with neighbors, participation in neighborhood association activities, and social trust and reciprocity index scores. Among the middle-aged group, it was significantly related to networking with neighbors and participation in neighborhood association activities but did not significantly predict the social trust and reciprocity index like the older group. When less of a misfit existed in terms of outdoor spaces, people were likely to network with their neighbors, participate more in neighborhood association activities, and have greater trust and reciprocity. These findings align with findings from Seoul, Korea [14] that suggested that the existence of and access to outdoor spaces such as green spaces and places to exercise were determinants of social capital. Moreover, outdoor spaces and buildings among AFCC's domain were consistently reported to correlate with better health and functional capacity [6]. Accessible rest areas and road conditions have also been shown to help with age-friendliness [6]. For social bonding, both outdoor spaces and transportation were important in a Korean AFCC study [29].

Transportation emerged as another important predictor of social capital. For the older group, transportation misfit index scores were significant for predicting social trust and reciprocity index scores. However, among the middle-aged group, transportation misfit index scores were significant in predicting participation in neighborhood association activities and social trust and reciprocity index scores. Hence, respondents of any age with low transportation misfit index scores were likely to have greater social trust and reciprocity

index scores. In addition, public transportation misfit was likely to affect participation in neighborhood association activities for the middle-aged group. The existing literature shows similar results. Providing affordable public transportation has been one of the main goals in many AFCCs [30]. Unavailable and inaccessible public transportation has indirect impacts on the quality of social participation [31] and active aging [8]. Limited transportation options are a barrier to many local communities making their environment more age-friendly [32,33] and create problems for older adults' mobility [34].

In the Korean context, local amenities for older adults (e.g., senior centers/clubs) and transportation to these amenities were important factors, as well as the road condition, in supporting active living [17]. Access to public transportation was still identified as an important determinant of social capital and resident walking activities in Seoul [35], even though many local communities in Korea provide frequent, inexpensive public transportation services [36]. Transportation was critical for low-income older adults as it was an important predictor of life satisfaction, providing more networking opportunities for social engagements and networking with their neighbors [37]. Given that over 40% of older adults lived under the poverty level in Korea in 2017 [38], many of them still sought more affordable transportation options. Interestingly, the availability of public transportation affected the types of activities older adults engaged in. For example, when older Korean adults lived in neighborhoods where convenient public transportation was available, supportive activities from clans and relatives were observed more often [15]. However, in walkable neighborhoods without public transportation, increased social support and networking were observed in neighborhood activity rooms (kyeong-ro-dang) [15].

The impact of a housing misfit on social capital was, surprisingly, less prevalent. The housing misfit was significant only for the older group in predicting their social trust and reciprocity index. Unlike transportation and outdoor spaces, the housing misfit had a positive relationship with social capital. When a high housing misfit existed, older adults were likely to have high social trust and reciprocity index scores. Older adults' perception of housing has been one of the key AFCC domains predicting health, functional ability, and quality of life [5–9]. As providing affordable housing and guidelines for accessible housing has been a primary concern for AFCC communities [30], it might be possible that there was more improvement of housing programs, and that older adults were stimulated to exchange help with neighbors and had to respect and trust their neighbors to seek better solutions and initiate efforts. Similar to our findings, in restructured urban neighborhoods in Dutch cities [4], local residents with a lower quality of life reportedly attended more neighborhood association activities to acquire and build social capital. In contrast, housing was not a significant indicator of the quality of life in a mid-Atlantic US AFCC [39]. One possible explanation for these different findings may be the different housing norms. Unlike in the United States, where single-family homes are preferred, many Koreans lived in multifamily housing communities. Residents living in multifamily unit communities, such as public housing, reported better relationships with neighbors [40] and interacted more within their neighborhood than their single-family-home counterparts [41]. Because half of the older adults in this study lived in multifamily home units, it is possible that they received more support from their neighbors and offered it in return. Thus, although a housing misfit emerged, their relationships with their neighbors helped to build trust and exchange supports. Among the AFCC NBE features, both the middle-aged and older groups were unaware of the availability of affordable and accessible housing options. More education and communication to promote awareness on these topics would be helpful for local residents.

The dwelling type affected all subcategories of social capital. Living in a dwelling other than a single-family home encouraged networking with neighbors, participating in neighborhood association activities, and social trust and reciprocity index for the older group; and participating in neighborhood association activities and the social trust and reciprocity index for the middle-aged group. Corner stores, plants, and outdoor seating space close to multifamily housing communities contributed to more networking with neighbors [18].

Multifamily home residents in Seoul interacted with their neighbors and built a social network by attending more community-associated meetings; as a result, overall, their relationships with neighbors improved, but they did not develop deeper trust with their neighbors [41]. Interestingly, among older adults living in low-rise housing, more social networking was observed with neighbors [42]. A popular gathering location was in front of the homes of residents with walking issues and, in addition to frequent interactions, they also exchanged help to overcome challenges as they became older and frailer [42].

The length of residence in one's current neighborhood was also positively correlated to networking with neighbors for both older and middle-aged groups. Specifically, among older adults, participating in neighborhood association activities was positively related to the length of residence, as were the trust and reciprocity index scores for middle-aged adults. However, previous studies [15] have shown negative relationships between the length of residence and social capital. For example, newcomers tended to interact more with their neighbors and attend neighborhood association activities in an effort to adapt to the new community [15]. Yet, in this study, the situation was reversed. The participants who had stayed longer in the current neighborhood were more likely to interact with their neighbors, attend neighborhood association activities, and have a high social trust and reciprocity index score.

Surprisingly, homeownership was significantly correlated to participating in neighborhood association activities only for the middle-aged group, and not significantly related to other aspects of social capital in either age group. However, homeownership was an important variable in a Dutch social capital study in a restructured urban neighborhood [4]. Being a homeowner was a determinant of participating in neighborhood association activities and volunteering opportunities [4]. Even though this was not significant in predicting social capital, homeownership was an important factor for NBE misfits for both age groups in Korea.

Finally, demographic characteristics mattered for social capital. While married, middle-aged participants were more likely to have greater social trust and reciprocity index scores, unmarried respondents (e.g., widowed, separate, single, etc.) in both groups reported more networking with neighbors. Education was a significant negative predictor for older adults' social trust and reciprocity index scores. Those with low education levels were likely to have high trust and reciprocity. Similarly, self-rated health status was a significantly negative predictor for networking with neighbors in the middle-aged group. Respondents in poorer health were more likely to report networking with neighbors. Demographic characteristics also mattered in other social capital studies in Korea. Women as well as less-educated and low-income individuals presented low participation levels in informal gatherings with family, friends, and neighbors and any public interest groups and any interactions to gain help and resources [43]. Similarly, women who were married, had higher education and income levels, and had good relationships with neighbors tended to have high reciprocity levels [44]. However, our study showed a negative relationship between women's education level and social capital, unlike in previous studies [43,44]. One contributing difference is that this study included adults 45 and older, while previous studies collected data from adults age 20 and over.

## 5. Conclusions

Unlike previous social capital studies in Korea that focused on larger urban cities, this study examined the role of neighborhood built environment (NBE) misfits in the two selected communities, utilizing existing age-friendly cities and communities approaches. Our findings in middle-aged and older adults confirmed that resident misfit with outdoor spaces and public transportation negatively impacts their social capital. Even though the person-environment misfit indices explained small amounts of variance in the models, their significance remains important to the respondents affected and should not be dismissed as their quality of life is likely lessened. The fit between the individual and their environment is critical to determining social connectivity and its mediated impacts on well-being [45],

and walkability is essential for building older adults' social capital [14,15,42]. To improve the neighborhood built environment, including walkability, will require not only infrastructure changes but a long-term commitment to policy change. In South Korea, where public policy at the local level is directly informed by policy decisions at the national level, more empirical studies are needed to demonstrate how age-friendly neighborhood built environment implementation influences the development of social capital, promotes health and well-being, and supports adherence to social norms, expectations, and participation in cultural activities. Such future activities will be better positioned [19] if they include cultural representatives, who can help guide the understanding of the role of NBE in the lives of older residents. They may also be able to refine the assessment measures to better address nuances in the NBE domains that may otherwise go unidentified. In particular, questions related to housing are needed to identify unmet needs, challenges, and gaps in housing types. Developing survey items that are specific to community infrastructure, environment, and culture may help advance our understanding of the built environment for future studies on age-friendly communities.

**Author Contributions:** E.H. analyzed the data and wrote the manuscript; N.B. conceptualized and revised the manuscript; J.W.J. and K.S. collected the data. All authors have read and agreed to the published version of the manuscript.

**Funding:** This research was funded by a National Research Foundation of Korea grant (NRF-2017S1A2A2041899) and the Virginia Tech Institute for Society, Culture and Environment and Institute for Critical Technology and Applied Science.

**Institutional Review Board Statement:** The study was conducted according to the policies and procedures guided by the Institutional Ethics Review Committee of Hanyang University (protocol code number: HYU-2019-03-005, approval date: 11 April 2019).

**Informed Consent Statement:** Informed consent was obtained from all subjects involved in the study.

**Conflicts of Interest:** The authors declare no conflict of interest.

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
