# Peer review of "The Impacts of the Neighborhood Built Environment on Social Capital for Middle-Aged and Elderly Koreans"

_sustainability, doi:10.3390/su13020756_

Round 1

Reviewer 1 Report

I would thank authors for this study of effect of neighbourhoods on the social capital among adults/old residents.

Several aspects could be considered to improve this paper:

  1. Introduction: the key research gap discussed in this part could be vague. Is it possible to apply more relevant literatures to enhance the discussion?
  2. Participants: it is unclear why the age threshold was set at 40 years and above. Apparently, there are big differences of neighbour attachments (social capitol) between the group (40 65) and the group (>65, retired persons; maybe 60 years in Korea). However, it is not clear why this study has put them together for the analysis. It seems that the analysis using this way might deliver some divergences from the way using two separate age groups.
  3. Methods: Is it possible to present more information of the two neighbourhoods. For example, locations, populations, building densities, etc.
  4. Methods:

It is unclear why different rating / assessment ways applied for various items of social capitals. For example, binary scale for Networking and Participation, while licker scale for social trust.

Minor aspects:

Table 1: the unit or meaning of the numbers inside could be clarified in the caption.

Table 2: more explanations of ‘Need help with daily activities’ (content, levels?); single family home = single family house? Dwelling type: what ‘others’ means? Homeownership: what ‘others’ means?

What the ‘accessible home’ means? This may need more clarification. 

Table4, typos: ‘B’ and ‘β’.

Author Response

On behalf of the team, we appreciate all the reviewers’ comments. The comments were very helpful revising the manuscript and we appreciate their thoughtful comments. Thank you for spending your time.  

Major changes:

There was a concern about the age threshold (40+) and describing/analyzing the data with “gender” differences. Instead of gender differences, we analyzed the social capital with “age” differences, the older (>65) and middle-aged (45-64) groups, as suggested by one of reviewers. Then, we re-ran statistical analysis and revised the content throughout the paper accordingly. Many parts were re-written. For example, we added more literature in Introduction, clarified Methods, added new tables, enhanced/extended Discussion and added conclusion/reflection in the end.   

Following are our responses to your specific comments.

Several aspects could be considered to improve this paper:

  1. Introduction: the key research gap discussed in this part could be vague. Is it possible to apply more relevant literatures to enhance the discussion?

     Author’s RESPONSE: Revised. Expanded/enhanced Introduction and Discussion.

  1. Participants: it is unclear why the age threshold was set at 40 years and above. Apparently, there are big differences of neighbour attachments (social capitol) between the group (40 65) and the group (>65, retired persons; maybe 60 years in Korea). However, it is not clear why this study has put them together for the analysis. It seems that the analysis using this way might deliver some divergences from the way using two separate age groups.

Author’s RESPONSE: Revised. We organized the paper with two different age groups: 1) 65+ (older group); 2) 45-64 (middle-aged group) throughout the paper and revised accordingly.

  1. Methods: Is it possible to present more information of the two neighbourhoods. For example, locations, populations, building densities, etc.

Author’s RESPONSE: Revised. Added more information of the two communities.

Lines 95-105.

  1. Methods: It is unclear why different rating / assessment ways applied for various items of social capitals. For example, binary scale for Networking and Participation, while licker scale for social trust.

Author’s RESPONSE: The binary coding was appropriate for the item that asked if interaction and participation occurred or didn’t, and the Likert –type scale was used on items measuring perceptions. Clarified this under 2.2. Measures and included separate tables per sub-category of social capital.

Minor aspects:

Table 1: the unit or meaning of the numbers inside could be clarified in the caption.

Author’s RESPONSE: Revised in Table 2 (NOTE: Table 1 from the previous version is revised in Table 2). Added “%” and helpful notes were added under Tables 5-7.

Table 2: more explanations of ‘Need help with daily activities’ (content, levels?); single family home = single family house? Dwelling type: what ‘others’ means? Homeownership: what ‘others’ means?

Author’s RESPONSE: Revised. Instead of using Need Help, self-rated health was used. Added examples of ‘others’ dwelling type in Table 1. We kept single family home as this has been used in housing studies in the United States.

What the ‘accessible home’ means? This may need more clarification. 

Author’s RESPONSE: Revised. Added examples under Housing in Table 2.

Table4, typos: ‘B’ and ‘β’ 

Author’s RESPONSE: Revised.

Reviewer 2 Report

The study takes into account the relation between the neighborhood built environment (NBE) aspects of age-friendly cities and communities (AFCC) on social capital in the Korean context.

The topic is very interesting and is treated appropriately. The paper is well-organized in the different sections (introduction, materials and methods, results and discussions). I would recommend adding some final reflections in conclusion to indicate future research developments.
The literature is summarized quite well in the text of the article and is referred to in a timely manner in the introduction and in the discussions.
In particular:
- Introduction
The introduction outlines and describes the state of affairs quite clearly. The authors have described, synthesized and commented on scientific literature. But is the analysis of the literature completely exhaustive? Is it perhaps necessary to introduce some other study? There are several studies that go into detail on the topics covered in this paper.
The objective of the study is outlined and described in an appropriate manner.
Lines 61-63: Perhaps I would move the sentence to the next section.
- Materials and method:
The method is clearly developed and is quite articulated in the different sections. In the different parts the steps are also described through exhaustive examples. The large amount of data and information, however, make some passages not very fluid in reading.
Some clarifications: table 1 shows part of the questionnaire already modified with respect to study 19? If so it could already be a result.
Are the results of the questionnaire related to this study shown in the columns Women and Men?
If so, I would at least move the results for the sample of people interviewed to the next section (results).
Results
Clear and appropriately described.
Line 172 Bivariate Results: attention the paragraph numbering is wrong.
Lines 189, 190: pay attention to the numbering of paragraphs and subparagraphs.
Reading the document may interest readers from different areas because the topic lends itself to being studied across the board involving different disciplines.

Author Response

On behalf of the team, we appreciate all the reviewers’ comments. The comments were very helpful revising the manuscript and we appreciate their thoughtful comments. Thank you for spending your time. 

Major changes:

There was a concern about the age threshold (40+) and describing/analyzing the data with “gender” differences. Instead of gender differences, we analyzed the social capital with “age” differences, the older (>65) and middle-aged (45-64) groups, as suggested by one of reviewers. Then, we re-ran statistical analysis and revised the content throughout the paper accordingly. Many parts were re-written. For example, we added more literature in Introduction, clarified Methods, added new tables, enhanced/extended Discussion and added conclusion/reflection in the end.  

Following are our responses to reviewers’ specific comments.

The topic is very interesting and is treated appropriately. The paper is well-organized in the different sections (introduction, materials and methods, results and discussions). I would recommend adding some final reflections in conclusion to indicate future research developments.
The literature is summarized quite well in the text of the article and is referred to in a timely manner in the introduction and in the discussions.

Author’s RESPONSE: Thank you. Revised. Added “5. Conclusion”

In particular:
- Introduction: The introduction outlines and describes the state of affairs quite clearly. The authors have described, synthesized and commented on scientific literature. But is the analysis of the literature completely exhaustive? Is it perhaps necessary to introduce some other study? There are several studies that go into detail on the topics covered in this paper. The objective of the study is outlined and described in an appropriate manner.

Author’s RESPONSE: Revised. Introduction’s expanded. Added related literature on the topic and briefly discussed a research gap.

Lines 61-63: Perhaps I would move the sentence to the next section.

Author’s RESPONSE: We considered this. However, after expanding the Introduction, it feels that the flow worked better keeping as they were.

- Materials and method:
The method is clearly developed and is quite articulated in the different sections. In the different parts the steps are also described through exhaustive examples. The large amount of data and information, however, make some passages not very fluid in reading. Some clarifications: table 1 shows part of the questionnaire already modified with respect to study 19? If so it could already be a result. Are the results of the questionnaire related to this study shown in the columns Women and Men? If so, I would at least move the results for the sample of people interviewed to the next section (results).

Author’s RESPONSE: Revised. Table 1 is Table 2 now and moved to “3. Results.”

-Results
Clear and appropriately described.
Line 172 Bivariate Results: attention the paragraph numbering is wrong.

Author’s RESPONSE: Thank you for catching this. We added “3.2.”, so 3.3. (Bivariate Results) is correct now.

Lines 189, 190: pay attention to the numbering of paragraphs and subparagraphs.
Reading the document may interest readers from different areas because the topic lends itself to being studied across the board involving different disciplines.

Author’s RESPONSE: Thank you for catching this. Revised. Numbering changed to “3.4.1.” and “3.4.2” now.

Reviewer 3 Report

The paper is interesting in its approach and conclusions. It could be important to indicate in the discussion the transferability of the method to other social contexts in order to validate the scientific method and the results.

Author Response

On behalf of the team, we appreciate all the reviewers’ comments. The comments were very helpful revising the manuscript and we appreciate their thoughtful comments. Thank you for spending your time. 

Major changes:

There was a concern about the age threshold (40+) and describing/analyzing the data with “gender” differences. Instead of gender differences, we analyzed the social capital with “age” differences, the older (>65) and middle-aged (45-64) groups, as suggested by one of reviewers. Then, we re-ran statistical analysis and revised the content throughout the paper accordingly. Many parts were re-written. For example, we added more literature in Introduction, clarified Methods, added new tables, enhanced/extended Discussion and added conclusion/reflection in the end.  

Reviewer 4 Report

The manuscript is interesting and relevant to the Journal. However, some major issues should be addressed according to the following general and specific comments/suggestions:

- Abstract: The method is not stated. Please include it briefly. In addition, add statistical data supporting results.

- Please remove Accessibility as keyword, a term more related to spatial analysis.

- Sentence 38-43 seems to be a contradiction. Please better justify the knowledge gap between AFFCC NBE's P-E and social capital.

- Please avoid using abbreviations when possible, they make the paper difficult to read.

- What does "# of Items" mean in Table 1?

- It is necessary to provide age-range information regarding participants' sample, section 2.1.

- Table 2. It seems that the mean age of women and men is relatively low and may affects results. Please include it as a limitation in that case.

- Line 154. Is the social capital the dependent variable in logistic and linear regression analysis? Please provide more information on the heuristic construct mentioned and used in the analysis.

- In general, some discussions points are too descriptive (they are presented as results) and there is a lack of contrast between them and the results of other references as well as the corresponding arguments. Please improve/reformulate this aspect/discussion points.

- Please add a conclusions section by separating the discussion form the conclusions (perhaps from line 353?). In the new conclusions section include strengths and limitations of the study.

Author Response

On behalf of the team, we appreciate all the reviewers’ comments. The comments were very helpful revising the manuscript and we appreciate their thoughtful comments. Thank you for spending your time. 

Major changes:

There was a concern about the age threshold (40+) and describing/analyzing the data with “gender” differences. Instead of gender differences, we analyzed the social capital with “age” differences, the older (>65) and middle-aged (45-64) groups, as suggested by one of reviewers. Then, we re-ran statistical analysis and revised the content throughout the paper accordingly. Many parts were re-written. For example, we added more literature in Introduction, clarified Methods, added new tables, enhanced/extended Discussion and added conclusion/reflection in the end.  

Following are our responses to reviewers’ specific comments.

- Abstract: The method is not stated. Please include it briefly. In addition, add statistical data supporting results.

Author’s RESPONSE: Revised. Added in lines 13-14.

- Please remove Accessibility as keyword, a term more related to spatial analysis.

Author’s RESPONSE: Revised. Removed “Accessibility” from line 24.

- Sentence 38-43 seems to be a contradiction. Please better justify the knowledge gap between AFFCC NBE's P-E and social capital.

Author’s RESPONSE: Revised. Introduction’s expanded. Attempted to address the P-E and social capital in lines 51-72.

- Please avoid using abbreviations when possible, they make the paper difficult to read.

Author’s RESPONSE: Revised. Tried to minimize using abbreviations and added captions for tables.

- What does "# of Items" mean in Table 1?

Author’s RESPONSE: Revised. Removed and put ranges in Table 2.

- It is necessary to provide age-range information regarding participants' sample, section 2.1.

Author’s RESPONSE: Revised. Added the age range in line 194 and Table 1.

- Table 2. It seems that the mean age of women and men is relatively low and may affects results. Please include it as a limitation in that case.

Author’s RESPONSE: Revised. Added “5. Conclusion.” Also changed the main analysis looking at age differences (65+ vs 45-64) instead of gender differences.

- Line 154. Is the social capital the dependent variable in logistic and linear regression analysis? Please provide more information on the heuristic construct mentioned and used in the analysis.

Author’s RESPONSE: Revised. Removed heuristic construct but for clarification, we put notes under tables and instead of putting one table with three sub-categories of social capital, we added a table per sub-category.

- In general, some discussions points are too descriptive (they are presented as results) and there is a lack of contrast between them and the results of other references as well as the corresponding arguments. Please improve/reformulate this aspect/discussion points.

Author’s RESPONSE: Revised. Expanded and enhanced “4. Discussion.” e.g., lines 364-413.

- Please add a conclusions section by separating the discussion form the conclusions (perhaps from line 353?). In the new conclusions section include strengths and limitations of the study.

Author’s RESPONSE: Revised. Added “5. Conclusion” with short reflections, strengths and limitations of the study

Reviewer 5 Report

This paper is well written and reveals an interesting topic.

This reviewer only present minor recommendations. Such as:

Abstract: This reviewer believes that the abstract may present the main values of the results and not only text.

L65. Sample and data collection?

L107. Start with "For the....". There's a dot before "Three", I believe it is a mistake, Or miss a paragraph

Results: It is possible to present effect sizes?

Author Response

On behalf of the team, we appreciate all the reviewers’ comments. The comments were very helpful revising the manuscript and we appreciate their thoughtful comments. Thank you for spending your time. 

Major changes:

There was a concern about the age threshold (40+) and describing/analyzing the data with “gender” differences. Instead of gender differences, we analyzed the social capital with “age” differences, the older (>65) and middle-aged (45-64) groups, as suggested by one of reviewers. Then, we re-ran statistical analysis and revised the content throughout the paper accordingly. Many parts were re-written. For example, we added more literature in Introduction, clarified Methods, added new tables, enhanced/extended Discussion and added conclusion/reflection in the end.  

Following are our responses to reviewers’ specific comments.

L65. Sample and data collection?

Author’s RESPONSE: Yes. Revised. Added “and Data Collection” in line 92. Thank you for catching this.

L107. Start with "For the....". There's a dot before "Three", I believe it is a mistake, Or miss a paragraph

Author’s RESPONSE: Revised. Added a space before three in line 146. Thank you for catching this.   

Results: It is possible to present effect sizes?

Author’s RESPONSE: Revised. New tables added. Instead of putting one table with three sub-categories of social capital (Tables 5-7), we added a table per sub-category and added confidence interval ranges and described model fit per category in texts.

Round 2

Reviewer 1 Report

My comments have been well addressed in the revision.

Author Response

Author’s RESPONSE: Your comments were very helpful revising the manuscript. Thank you. 

Reviewer 2 Report

It seems to me that the paper thus modified is more correct and above all more complete. I really appreciated the review of the "materials and methods" section made by the authors; this change makes the whole paper more fluid and understandable. The addition of bibliographic references and some reflections completes the work in a much more exhaustive way than the previous version of the article.

Author Response

Author’s RESPONSE: Your comments were very helpful revising the manuscript. Thank you. An English editing service was requested and completed before the second revision. 

Reviewer 4 Report

The manuscript was clearly improved. However, some minor issues should be addressed as follows.

Please add statistical data to the results included in the abstract of your manuscript.

Conclusions section, line 462. Please remove the terms "small and medium-sized cities" as this is the first time they are show throughout the manuscript.

Please add reference/es to support this statement (lines 82-84).

Author Response

The manuscript was clearly improved. However, some minor issues should be addressed as follows.

Please add statistical data to the results included in the abstract of your manuscript.

Author’s RESPONSE: Revised. Contents added to lines 211-214, 278-283, and 319-320.

Conclusions section, line 462. Please remove the terms "small and medium-sized cities" as this is the first time they are show throughout the manuscript.

Author’s RESPONSE: Revised. Removed the terms from line 464 and inserted “two selected communities” in line 470 instead.

Please add reference/es to support this statement (lines 82-84).

Author’s RESPONSE: Revised. Added references “11” and “14” in line 86.

Your comments were very helpful revising the manuscript. Thank you.